# Demethoxycurcumin-Loaded Chitosan Nanoparticle Downregulates DNA Repair Pathway to Improve Cisplatin-Induced Apoptosis in Non-Small Cell Lung Cancer

**DOI:** 10.3390/molecules23123217

**Published:** 2018-12-05

**Authors:** Ying-Yi Chen, Yu-Jung Lin, Wei-Ting Huang, Chin-Chuan Hung, Hui-Yi Lin, Yu-Chen Tu, Dean-Mo Liu, Shou-Jen Lan, Ming-Jyh Sheu

**Affiliations:** 1School of Pharmacy, China Medical University, Hsueh-Hsih Road, Taichung 40402, Taiwan; y200260@yahoo.com.tw (Y.-Y.C.); cc0206hung@gmail.com (C.-C.H.); hylin@mail.cmu.edu.tw (H.-Y.L.); icelemon0215@gmail.com (Y.-C.T.); 2Department of Pharmacy, Chang Bing Show Chwan Memorial Hospital, No.6, Lugong Rd. Lugang Town, Changhua County 505, Taiwan; muroa0412@gmail.com; 3Department of Materials Science and Engineering, National Chiao Tung University, 1001 University Road, Hsinchu 300, Taiwan; shinichik76@gmail.com (W.-T.H.); deanmo.liu@gmail.com (D.-M.L.); 4Department of Healthcare Administration, Asia University, Taichung 41354, Taiwan; sjlan@asia.edu.tw

**Keywords:** demethoxycurcumin, chitosan, ERCC1, NSCLC

## Abstract

Demethoxycurcumin (DMC), through a self-assembled amphiphilic carbomethyl-hexanoyl chitosan (CHC) nanomatrix has been successfully developed and used as a therapeutic approach to inhibit cisplatin-induced drug resistance by suppressing excision repair cross-complementary 1 (ERCC1) in non-small cell lung carcinoma cells (NSCLC). Previously, DMC significantly inhibited on-target cisplatin resistance protein, ERCC1, via PI3K-Akt-snail pathways in NSCLC. However, low water solubility and bioavailability of DMC causes systemic elimination and prevents its clinical application. To increase its bioavailability and targeting capacity toward cancer cells, a DMC-polyvinylpyrrolidone core phase was prepared, followed by encapsulating in a CHC shell to form a DMC-loaded core-shell hydrogel nanoparticles (DMC-CHC NPs). We aimed to understand whether DMC-CHC NPs efficiently potentiate cisplatin-induced apoptosis through downregulation of ERCC1 in NSCLC. DMC-CHC NPs displayed good cellular uptake efficiency. Dissolved in water, DMC-CHC NPs showed comparable cytotoxic potency with free DMC (dissolved in DMSO). A sulforhodamine B (SRB) assay indicated that DMC-CHC NPs significantly increased cisplatin-induced cytotoxicity by highly efficient intracellular delivery of the encapsulated DMC. A combination of DMC-CHC NPs and cisplatin significantly inhibited on-target cisplatin resistance protein, ERCC1, via the PI3K-Akt pathway. Also, this combination treatment markedly increased the post-target cisplatin resistance pathway including bax, and cytochrome c expressions. Thymidine phosphorylase (TP), a main role of the pyrimidine salvage pathway, was also highly inhibited by the combination treatment. The results suggested that enhancement of the cytotoxicity to cisplatin via administration of DMC-CHC NPs was mediated by down-regulation of the expression of TP, and ERCC1, regulated via the PI3K-Akt pathway.

## 1. Introduction

Clinically, cisplatin-based chemotherapy is the first-line treatment for non-small-cell lung cancer (NSCLC). By forming the DNA adducts, cisplatin has been shown to be effective against cytotoxic effects and inhibit the DNA replication. Nevertheless, drug resistance develops and reduces the clinical efficacy of cisplatin in NSCLC [1]. The development of resistance to cisplatin has been characterized by an increased tolerance where damaged DNA is repaired by the nucleotide excision repair (NER) pathway [2]. During the DNA damage of the cancer cells, the NER pathway becomes responsible for repairing bulky lesions [3]. Consequently, the NER pathway plays important roles by removing cisplatin-induced adducts from the DNA. Several steps participate in cisplatin-resistance of tumor cells. First, the damaged DNA is documented and bound by a protein complex called xeroderma pigmentosum A (XPA), which is the rate-limiting protein in the process of NER initiation. Second, transcription factor II (TFIIH) and xeroderma pigmentosum complementation group D (XPD) unwind the DNA. Third, excision repair cross-complementation 1 (ERCC1) plays a major role in the incision at the 5′ site of damaged DNA [4]. Finally, normal nucleotide sequences are restored by the DNA polymerases [5]. Overexpression of ERCC1 in cancer cells clears the cisplatin-induced DNA adducts, and brings about the drug resistance [4]. Despite the controversy, researchers focused on understanding roles of ERCC1 in sensitizing the cisplatin-based chemotherapy for lung cancer. It has been described that the patients with metastatic lung cancer under the treatment of platinum-based chemotherapy benefit from a lower level of ERCC1 [6]. Therefore, ERCC1 is considered to be one of the significant therapeutic target for lung cancer treatment, as targeting ERCC1 may possibly restore the therapeutic sensitivity to platinum-based chemicals [7].

Thymidine phosphorylase (TP) is an enzyme essential for the pyrimidine salvage pathway. It is particularly upregulated in several solid tumors. An increased level of TP is correlated with simultaneous overexpression of other angiogenic factors (i.e., matrix metalloproteases, interleukins, and vascular endothelial growth factor), causing development of angiogenesis and cancer metastasis. Moreover, exceeding TP level shelters tumor cells from apoptosis [8,9]. In clinics, patients found with TP overexpression show poor prognosis and shorter overall survival rate [10]. Thus, TP is recognized as a key target for developing novel anticancer agents. 

Curcuminoids are those polyphenol coloring compounds originating from *Curcuma longa* Linn. It contains three major bioactive ingredients—curcumin, demethoxycurcumin (DMC), and bisdemethoxycurcumin (BDMC)—in a ratio of 77:17:3 [11]. Previous research pointed out that coadministration with curcumin and cisplatin on cancer cells increased cytotoxicity to cisplatin, and was mediated by down-regulation of the expression levels of TP and ERCC1 and by inactivation of ERK1/2 [12]. Our previous reports suggested that enhancement of the cytotoxicity to cisplatin by coadministration with DMC was mediated by down-regulation of the expression of TP and ERCC1, regulated via the PI3K-Akt-Snail pathway [13]. Compared to other curcuminoids, DMC showed the most potent inhibition of ERCC1 from cisplatin treatment; however, the low water solubility, poor gastrointestinal absorption, low bioavailability, rapid metabolism, and systemic elimination prevents its clinical application [14]. To overcome the limitations, many researchers have engaged diverse approaches to improve the absorption and bioavailability of curcuminoids [15]. In our recent work, formation of a DMC nanocrystallite-chitosan nanocarrier for controlled low dose cellular release would be valuable for further application.

In this work, the controlled release of DMC from the nanocrystallite-chitosan nanocarrier has been examined for its possibility to enhance cisplatin-induced apoptosis by downregulation of TP and ERCC1-related pathways in NSCLC. In order to accurately regulate the DMC elution, a highly biocompatible amphiphilic chitosan was employed as a drug carrier [16,17]. This amphiphilic carboxymethyl-hexanoyl chitosan (CHC) is modified from natural chitosan through carboxylmethylation and hexanoyl replacement along the backbone of pristine chitosan. This modified chitosan has been evidenced to be highly dissoluble in an aqueous solution of neutral pH, is biocompatible, and can self-assemble to form well-defined nanocapsules in many water-solvent mixtures [17]. This study was focused on the possibility that DMC-CHC NPs potentiated chemotherapy with cisplatin and its correlation with TP and ERCC1 signaling pathways.

## 2. Results

### 2.1. Characterization of the Characteristics of Unloaded CHC Nanoparticles (such as Hydrodynamic Radius (D_h_), Zeta-Potential, TEM/SEM Morphology), and to Compare with that of DMC-CHC Nanoparticles (DMC-CHC NPs)

According our previous study, the characterization of CHC and DMC-CHC NPs were described [18,19]. The loading efficiency of DMC into the nanocarrier CHC was determined to be 98% using HPLC. Furthermore, the drug encapsulation efficiency was 16.4%. The morphology and size of DMC-CHC NPs was analyzed using transmission electron microscopy (TEM). TEM analysis revealed an increase in size for DMC-CHC NPs, with the DMC being randomly distributed as nanocrystals through the CHC phase (seen as black spots in the Figure 1A). This finding suggested the DMC molecules being crystallized and distributed randomly throughout the molecular framework of the CHC nanomatrix. Furthermore, the hydrodynamic diameter from DLS and zeta potential of different formulations in pH 7.4 PBS were determined (Table 1). 

### 2.2. In Vitro Cytotoxicity 

Due to the low water solubility, poor gastrointestinal absorption, and low bioavailability, DMC limits itself for clinical application. Therefore, forming of DMC nanocrystallite-chitosan nanocarrier for controlled low dose cellular release was carefully considered to solve the problems. Subsequently, the cytotoxic effects of DMC-CHC NPs (dissolved in ddH_2_O) was compared with free DMC (dissolved in DMSO) on A549 cells. Being pretreated with various doses (0, 6.25, 12.5, 25, 50, and 100 μg/mL) of DMC and DMC-CHC NPs for 48 h on A549, the IC_50_ shown for the free DMC and the DMC-CHC NPs were 18.1 ± 0.3 and 27.1 ± 0.1 μg/mL, respectively (Figure 1B). Our results suggested that DMC-CHC NPs could be further applied providing DMC showed rapid cellular internalization and was directly released into the cell as previous described [17]. To determine whether CHC could serve as a potential candidate for DMC delivery applications, CHC nanomatrixes (6.25, 12.5, 25, 50, and 100 μg/mL) were incubated with A549 for 24 h, 48 h, and 72 h. The results showed that the CHC nanomatrixes demonstrated no cytotoxicity on A549 (Figure 1C). Therefore, CHC nanomaterials was selected for drug delivery applications. 

### 2.3. Fluorescence Confocal Microscopy 

To understand its intracellular dynamics, entrapped fluorescein isothiocyanate (FITC)-labeled DMC-CHC NPs in A549 cells were observed using laser scanning confocal microscopy. These merged images of the FITC channel (DMC-CHC NPs, green), 4′,6-diamidino-2-phenylindole (DAPI) channel (nuclei, blue), and rhodamine channel (F-actin, red) are shown (Figure 2A). When treated with FITC-DMC-CHC NPs (20 μg/mL) for 1, 2, and 4 h, the FITC-DMC-CHC NPs (green fluorescence spectrum) were monitored and clearly found inside of A549 cells. Our results confirmed that the FITC-DMC-CHC NPs were internalized by A549 in 1 h of treatment, and found in a punctate pattern around the nuclei (Figure 2A). The results approved that the DMC-CHC NPs were mostly and efficiently taken up by A549 cells. 

### 2.4. Cellular Uptake 

In order to quantify the cellular uptake of the DMC-CHC NPs in A549, the fluorescence intensity of A549 after incubation with the FITC-CHC and FITC-DMC-CHC NPs was determined using a flow cytometry analyzer. Being exposed in A549 for 8 h, the green fluorescence intensity from FITC-CHC NPs (20 μg/mL) increased in a time-dependent manner (Figure 2B); also, similar performance has been presented with the FITC-DMC-CHC NPs (5 μg/mL DMC) (Figure 2B). For different time periods (0, 5 h, 1 h, 2 h, and 4 h), the proportional amount of the FITC-CHC NPs taken up by A549 (M1) was relatively similar. FITC-DMC-CHC NPs have also been internalized into A549 cells in an efficient way, demonstrating the accumulated NPs for 0.5 h, 1 h, 2 h, and 4 h, corresponding to a similar value (Figure 2B). By analyzing 10,000 events, the fluorescence intensity of FITC-CHC NPs exhibited an equal value compared with FITC-DMC-CHC NPs. This information verified a comparable internalization of the FITC-CHC NPs by A549 cells and that of FITC-DMC-CHC NPs (Figure 2B).

### 2.5. DMC-CHC NPs Further Decreased Cisplatin-Induced Cell Viability

DMC-CHC NPs was conducted to investigate whether it acted synergistically with cisplatin-induced lung cancer cells’ viability. A549 were coadministered with cisplatin (0.0125, 0.125, 1.25, 12.5, 25, and 50 M) and various doses of DMC-CHC NPs (2.5, 5.0, 10.0, and 15 g/mL) for 72 h. Our results showed that DMC-CHC NPs acted synergistically with cisplatin, inducing cell cytotoxic activity (Figure 3).

### 2.6. DMC-CHC NPs Reduced TP and ERCC1 Protein Expressions in Cisplatin-Treated Cell

We wanted to understand whether a combination treatment with DMC-CHC NPs and cisplatin significantly reduced TP and ERCC1 overexpression compared to cisplatin treatment alone. A549 were exposed on cisplatin alone or with various doses (7.5, 15, and 30 g/mL) of DMC-CHC NPs for 48 h. In the presence of DMC-CHC NPs, our results showed that this combination treatment significantly decreased both TP and ERCC1 protein expressions when compared to the cisplatin control group (Figure 4A).

### 2.7. DMC-CHC NPs Decreased Protein Levels of PI3K, and Phosphorylation AKT Induced by Cisplatin

Subsequently, to assess the possible TP, ERCC1, and ERCC1 related molecular mechanisms, which occur upon cotreatment with DMC-CHC NPs and cisplatin, the protein levels of snail, PI3K, and phosphorylation AKT were examined. DMC-CHC NPs decreased the protein levels of snail, PI3K, and phosphorylation AKT in cisplatin-treated A549 cells. These results show that DMC-CHC NPs down-regulates cisplatin-induced PI3K and phospho-AKT activation along with the expression of TP and ERCC1 (Figure 4B).

### 2.8. DMC-CHC NPs Altered Protein Levels of Bax, and Cyt c Induced by Cisplatin

The combined effects of DMC-CHC NPs and cisplatin on the expression of bax, bcl-2, cyt *c*, and bax/bcl-2 were conducted to explore its synergistic apoptotic effect. Figure 4B shows that cisplatin decreased the levels of bax, and cyt *c*, and DMC-CHC NPs markedly increased bax, the bax/bcl-2 ratio, and cyt *c* protein expressions.

## 3. Discussion

Previously, CHC-based nanocarriers have been established as having superb biocompatibility [17,20] and that they encapsulate and deliver DMC with high efficacy [17]. Our goal was to clarify how DMC-loaded CHC nanocarriers could efficiently inhibit the cisplatin-induced ERCC1 overexpression, which accounts for the development of resistance in NSCLC. Wang et al. suggested that this modified amphiphilic chitosan has confirmed the capacity to self-associate in contact with aqueous media to form polymeric nanoparticles and hydrogels [18,21]. Importantly, these structures can be successfully applied for the encapsulation and the release of DMC. Our data suggested that DMC showed the most potent inhibition of A549 cell viability. When dissolved in water, DMC-CHC NPs showed comparable cytotoxic potency with free DMC (dissolved in DMSO) (Figure 1B). Confocal images showed that the DMC-CHC NPs were efficiently taken up by A549 cells (Figure 2A). Cell uptake efficiency was measured using flow cytometry analysis for FITC-CHC (25 μg/mL) and FITC-DMC-CHC nanoparticles (5 μg/mL) in A549 cells (Figure 2B). A combination treatment with DMC-CHC NPs and cisplatin significantly decreased the cell viability compared to the cisplatin alone treatment (Figure 3). The molecular mechanisms indicated that DMC-CHC NPs could modulate the cisplatin-induced resistance of the cancer cells via both on- and post-target resistance signaling (Figure 4A,B). 

The size of the intracellular vesicles comprising the endocytosed nanoparticles provide the main factor concerning the uptake mechanism [22]. Allowing the cell surface to identify and internalize through the cellular pathways depends on two significant factors including particle size and shape [23]. The size of the CHC nanomatrix (about 150 nm) is similar to DMC-CHC NPs. Thus, DMC-CHC NPs could be proficiently internalized by A549 cells, still holding an adequate dose of DMC to successfully inhibit cisplatin-induced ERCC1 overexpression. In our previous study, the data showed that free DMC was completely eluted in 24 h, while DMC-CHC NPs showed a sustained, slow profile for a time period of 10 days (corresponding to an accumulative amount of about 65%). In the present study, for 48 h and 72 h incubation periods, this evidence shows that the DMC, gradually slow-released from the nanoparticle, could effectively work on the cisplatin-induced resistance

Several molecular mechanisms account for the cisplatin resistance. These comprise of the efflux of medications, cellular uptake of the drugs, inhibition of cancer apoptosis, and enhanced DNA repair of cancer cells [4]. Particularly, the DNA repair ability is frequently raised to dodge the platinum-induced death of cancer cells. Our results indicated that DMC-CHC NPs decreased the ERCC1 over-expression (Figure 4A) that is often linked with chemoresistance [24]. It has been found that the enhancement of the cytotoxicity to cisplatin via the administration of curcuminoids is mediated by down-regulation of the expression levels of TP and ERCC1 [12,13]. Additionally, curcumin-mediated inhibition of the Fanconi anemia/BRCA pathway sensitizes tumor cells to cisplatin, resulting in apoptotic death of ovarian and breast tumor cell lines [25]. In the present study, we have shown that DMC-CHC NPs stimulates sensitivity of cisplatin to NSCLC via TP and ERCC1 down-regulation (Figure 4A). Researchers document changes of ERCC1 that contribute to this exceptional DNA repair signaling in cisplatin-resistant cancer cells that may actually make it possible to develop novel therapies for cisplatin-resistant lung cancer cancers [26].

Many pathways may be responsible for the change of ERCC1 in cancer cells, including lung cancers. Some results suggest that twist-related protein 1 (TWIST1)- and epithelial–mesenchymal transition (EMT)-driven increases in Akt activation, and thus tumour cell proliferation, as a potential mechanism of drug resistance in epithelial ovarian cancer [27]. PI3K-Akt-Snail-ERCC1 pathway attributes to the development of resistance to cisplatin chemotherapy, and inhibition of PI3K/Akt/ERCC1 pathways brought about a substantial decrease of resistance to cisplatin treatment [28]. These results advise that down-regulation of ERCC1 leads to apoptosis in A549 cells. Moreover, DMC-CHC NPs apparently reduced the phosphorylation AKT protein level in A549 cells (Figure 4A). Our study demonstrated that the PI3K-AKT-ERCC1 pathway signaling molecules was also remarkably reduced after DMC-CHC NPs treatment (Figure 4A). These results made available evidence of molecular mechanisms to support the findings that DMC-CHC NPs had inhibitory effects on the PI3K-AKT-ERCC1 pathway signaling in A549 cells (Figure 4A). Regarding the post-target cisplatin resistance mechanisms, our study indicated that DMC-CHC NPs could alter the bax and cytochrome *c* signaling, which is responsible the cancer cell apoptosis. The DMC-CHC NPs significantly increased bax and cytochrome *c* in a dose-dependent manner (Figure 4B). Therefore, this could explain how the DMC-CHC NPs combined with cisplatin could decrease the cancer cell viability (Figure 3). 

Although DMC and BDMC show similar structures with curcumin, yet the are less-contained in and harder-to-be-separated from *Curcuma longa*, which limit their studies. Recently, our collaborator efficiently isolated and characterized curcumin, DMC, and BDMC from *Curcuma longa,* and the purity present was over 98% [29]. DMC demonstrated the most potent inhibition on balloon injury-induced vascular smooth muscle cells (VSMCs) migration and neointima formation when compared to other curcuminoids [30]. In rhodamine 123 efflux and calcein-AM accumulation assays, DMC exhibited the highest inhibition potency among DMC, curcumin, BDMC, and other assays [31]. Furthermore, DMC demonstrated the most efficient cytotoxic effects on prostate cancer PC3 cells via AMP-activated protein kinase (AMPK)-induced down-regulation of heat-shock protein (HSP) 70 and epidermal growth factor receptor (EGFR) [29]. In the present study, DMC-CHC NPs demonstrated comparable cytotoxic effects compared to free DMC in A549 (Figure 1B), and decrease cisplatin-induced cell viability (Figure 4) through downregulation of TP and ERCC1-related pathways (Figure 4A,B) in NSCLC.

## 4. Materials and Methods

### 4.1. Agents

3-(4,5-Dimethylthiazol-2-yl)-2,5-diphenyltetrazolium bromide (MTT), dimethyl Sulfoxide (DMSO), fluorescein isothiocyanate (FITC), antibody for β-actin, and phosphate buffered saline (PBS) were purchased from Sigma (St. Louis, MO, USA). Demethoxycurcumin and DMC-CHC NPs were kindly sponsored by Dr. Hui-Yi Lin (School of Pharmacy, China Medical University, Taichung, Taiwan), and Dr. Dean-Mo Liu (Nano-Bioengineering Lab, Department of Material Science and Engineering, BioICT Consortium, National Chiao Tung University, Hsinch, Taiwan), respectively. Our data indicated that the purity of this DMC had a purity of at least 99%. Antibodies for ERCC1, TP, phospho-Akt, PI3K, bax, and cyt *c* were obtained from Cell Signaling Technology (Beverly, MA, USA). Antibodies for mouse and rabbit conjugated with horseradish peroxidase (HRP) were purchased from Chemicon (Temecula, CA, USA). Western chemiluminescent HRP substrate was obtained from Millipore Corp. (Billerica, MA, USA). 

### 4.2. Characterization of CHC and Preparation of DMC-CHC NPs

According to our previous report [18], DMC-loaded self-assembled CHC nanoparticles were prepared as followed: 5 mg CHC in 8 mL H_2_O was mixed with 2 mg of DMC in 2 mL methanol and stirred up for 12 h at room temperature. The methanol was then removed using a rotary evaporator (BÜCHI, Switzerland). Non-loaded DMC was removed via centrifugation at 12,000 rpm for 30 min, discarding the pellet. An amphiphilic chemically-modified chitosan, carboxymethyl-hexanoyl chitosan (CHC), was successfully synthesized as previously described [32] and was purchased from (Advanced Delivery Technologies, Inc. Hsinchu, Taiwan) and used without further purification. The characterizations of the CHC has been shown as followed: degree of deacylation of 91.7%, average molecular weight of 3.5 × 10^4^ g/mol (with dispersity of 1.12), and degree of carboxymethylation of 55.3%. The average molecular weight was determined using GPC (TSKgel SuperAW4000 Column; PU-4180 RHPLC Pump, RI-4030 Refractive Index Detector, Hitachi, Tokyo, Japan, LC-NetII/ADC Interface Box and CO-4060 Column Oven, Hitachi, Tokyo, Japan) using polyethylene (PEO/PEG) as a standard and the degree of carboxymethylation was determined using FT-IR.

### 4.3. TEM Image

TEM image of the nanoparticles have been reported previously [18]. The structure of CHC and DMC-CHC NPs were analyzed using transmission electron microscopy (TEM, JEOL 2100, Tokyo, Japan). TEM samples were prepared immediately after the CHC and DMC-CHC NPs had been synthesized. Both samples (1 mg/mL) were put on 200-mesh copper grids coated with carbon. For the DMC-CHC NPs sample, negative staining was used to observe DMC in the CHC carrier.

### 4.4. Determination of the Loading Efficiency and the Drug Encapsulation Efficiency

The loading efficiency was determined by analyzing the amount of non-loaded DMC using high performance liquid chromatography (HPLC) with a C18 column, monitoring absorbance at 425 nm. The drug-loading efficiency (DLE, %) was calculated using the following equation: DLE (%) = [(amount of DMC in nanoparticles)/(amount of DMC used for nanoparticle preparation)] × 100. Furthermore, the drug encapsulation efficiency was determined using the following formula: DEE (%) = [(mass of DMC in nanoparticles)/(mass of DMC-CHC nanoparticles)] × 100.

### 4.5. Cells and Cell Culture

Human lung bronchioloalveolar adenocarcinoma A549 cells were obtained from the Food Industry Research and Development Institute (Hsinchu, Taiwan). A549 cells were grown in F-12 Nutrient Mixture (F-12; Gibco), supplemented with 10% fetal bovine serum (FBS, Gibco), 100 units/mL penicillin G, and 100 mg/mL streptomycin sulfates at 37 °C in a humidified atmosphere of 5% CO_2_. The culture medium was replaced every two days and cells were passaged twice every week.

### 4.6. Fluorescence Confocal Microscopy

A549 cells (2 × 10^3^ cells) were cultured on coverslips and incubated with or without FITC-DMC-CHC NPs (10 μg/mL) for 0, 1, 2, and 4 h. Thereafter, A549 cells were fixed with 3.7% formaldehyde and permeabilized with 0.1% Triton X-100. Then, the cells were washed twice with PBS and incubated at 24 °C with rhodamine-phalloidin overnight. The cells were washed with PBS, and stained with DAPI for 1 h. Furthermore, the coverslips were mounted on glass slides using a mounting solution (Dako, Santa Clara, CA, United States) and analyzed at multiple focal planes using a Leica TCS SP2 Laser Scanning Confocal Microscope (Heidelberg, Germany). 

### 4.7. Cellular Uptake

Internalization of FITC-CHC NPs and FITC-DMC-CHC NPs with the A549 cells was determined using a flow cytometer. A549 (1 × 10^4^ cells) cells were plated in a 6 cm dish and allowed to attach for 24 h. The culture medium was discarded, and the remaining cells were washed with PBS. To determine the time-dependent cellular uptake of the NPs effect, the cells were incubated with FITC-CHC NPs and FITC-DMC-CHC NPs for 0, 1, 2, and 4 h at 37 °C. Then, A549 cells were washed twice with PBS and harvested using trypsinization. After that, A549 cells were centrifuged and collected for further dehydration with 70% ethanol (−20 °C, overnight) to fix the cells. Finally, this process was resuspended with PBS. The cell solutions were filtered through a nylon membrane (BD Biosciences, San Jose, CA, USA) to avoid cell aggregation. To determine the cellular uptake of nanoparticles, ten thousand cells were analyzed using BD FACSCalibur flow cytometry, and the fluorescence intensity was quantified using CellQuest software CellQuest Pro software (version 6.1.2, BD Biosciences, USA).

### 4.8. Cell Viability Assay (MTT Assay)

A549 cells (2 × 10^4^ cells/well) were seeded into a 96-well plates overnight. A549 cells were exposed to 100 μL of different concentrations of the tested drugs (e.g., EEAC or paclitaxel) in a culture medium. After 24 h of treatment, 10 μL of 5 mg/mL MTT (3-(4,5-dimethylthiazol-2-yl)-2,5-diphenyl tetrazolium bromide) was added into each well. After 4 h of incubation, cells were washed twice with 1× PBS, and then 200 μL of dimethyl sulfoxide (DMSO) was added to each well. Absorbance values at 570 nm were determined for each well using 650 nm as the reference wavelength. The absorbance could be correlated to the percentage of vital cells via comparison with the control group (without treatment of tested drugs). The cell viability ratio was calculated using the following formula: cell viability (%) = OD (treated)/OD (control) × 100%.

### 4.9. Sulforhodamine B Assay

The sulforhodamine B (SRB) assay used for the cell density determination was based on the measurement of the cellular protein content. Cells were grown in 96-well plates and coated in gelatin for 30 min. After 30 min, the cells were incubated with DMC-CHC NPs (2.5, 5.0, 10.0, and 15.0 μg/mL) and various doses of cisplatin (2.5, 5.0, 10.0, and 15.0 μg/mL), and incubated in a humidified 5% CO_2_ atmosphere at 37 °C for 72 h. The cells were added for 30 min with 50% trichloroacetic acid (TCA) 25 μL/well and subsequently washed twice time with deionized distilled water (DDW). The plates were air-dried and stained for 30 min with 0.04% SRB 50 μL/well and subsequently washed twice with 1% acetic acid to remove unbound stains. The plates were air-dried and bound protein stain was solubilized with Tris base 100 μL/well and the absorbance at 515 nm was measured for each well on an ELISA reader.

### 4.10. Western Blotting Analysis

Following the same method as our previous study [33], A549 cells were plated in 10-cm dishes (2 × 10^6^ cells/well) and incubated with cisplatin (25 μM) and/or 7.5, 15.0, and 30.0 μg/mL of DMC-CHC NPs in F-12 containing 1% FBS for 48 h. The cells were collected, then lysed in a lysis solution (iNtRON, Seongnam, Korea.), followed by incubation at 95 °C for 5 min. Samples were separated in a 10% sodium dodecyl sulfate polyacrylamide gel electrophoresis (SDS–PAGE) gel and then transferred onto a polyvinylidene difluoride (PVDF) membrane. The membrane was blocked in 5% non-fat milk in a PBS-Tween 20 buffer for 1 h and probed with antibodies specific for bax, bcl2, cyt *c*, PARP, ERCC1, TP, PI3K, phospho-AKT, and snail overnight at 4 °C. The blots were then incubated with horseradish peroxidase-linked secondary antibody for 1 h followed by development with the electrochemiluminsence (ECL) reagent (Millipore, Bedford, MA, USA) and exposed to LAS-4000 (Fujifilm, Fujifilm LAS-4000 system (San Leandro, CA, USA). The data were analyzed using a Multi Gauge Imaging Systems, Molecular Imaging Software (version 3.0, Fujifilm, San Leandro, CA, USA).

### 4.11. Statistics Analysis

The data was compared between groups of cytotoxicity assays and Western blotting analysis using one-way analysis of variance. All other data were normally distributed, and therefore Student’s *t*-test was used. A value of *p* < 0.05 was considered statistically significant.

## 5. Conclusions

A novel pharmaceutical nanoformulation based on DMC-loaded amphiphilic chitosan nanomatrix was designed to investigate the cellular internalization and inhibitory effect on cisplatin-induced TP and ERCC1 overexpression in NSCLC. Compared to free DMC, DMC-CHC NPs showed comparable potency and a tremendous inhibition influence on cisplatin-induced TP and ERCC1 overexpression via a low-dose elution profile. This nanoparticle could provide us more insight of the future success of the study to overcome cisplatin-caused drug resistance. 

## Figures and Tables

**Figure 1 molecules-23-03217-f001:**
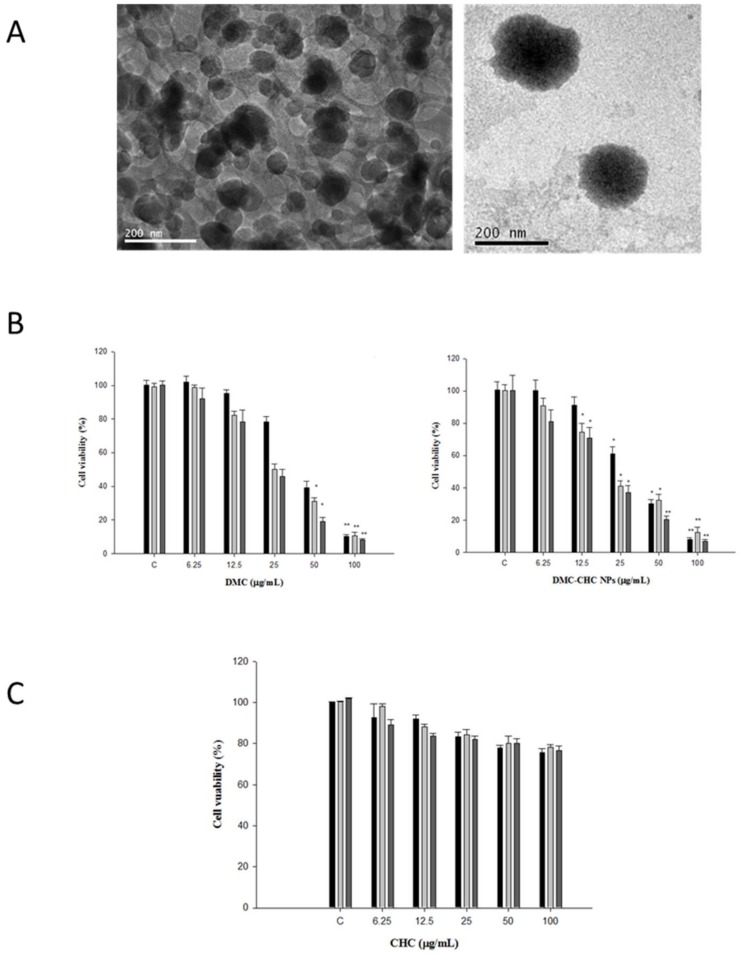
Characteristics of CHC and DMC-CHC NPs. (**A**) TEM morphology of CHC and DMC-CHC NPs (scale bar: 200 nm). Left one is unloaded CHC, and right one is DMC-CHC nanoparticles. (**B**) Left panel: effect of DMC on the viability of A549 cells. Right panel: effect of free DMC-CHC NPs on the viability of A549 cells. (**C**) CHC nanomatrix (6.25, 12.5, 25, 50, and 100 μg/mL) were incubated with A549 for 24 h, 48 h, and 72 h. The results showed that the CHC nanomatrixes demonstrated no cytotoxicity on A549. Each data point is represented as the mean ± SD (*n* = 3). Error bars represent standard deviation. * Indicates the values significantly different from the control time points, 24 h, 48 h, and 72 h., respectively. (* *p* < 0.05; ** *p* < 0.01).

**Figure 2 molecules-23-03217-f002:**
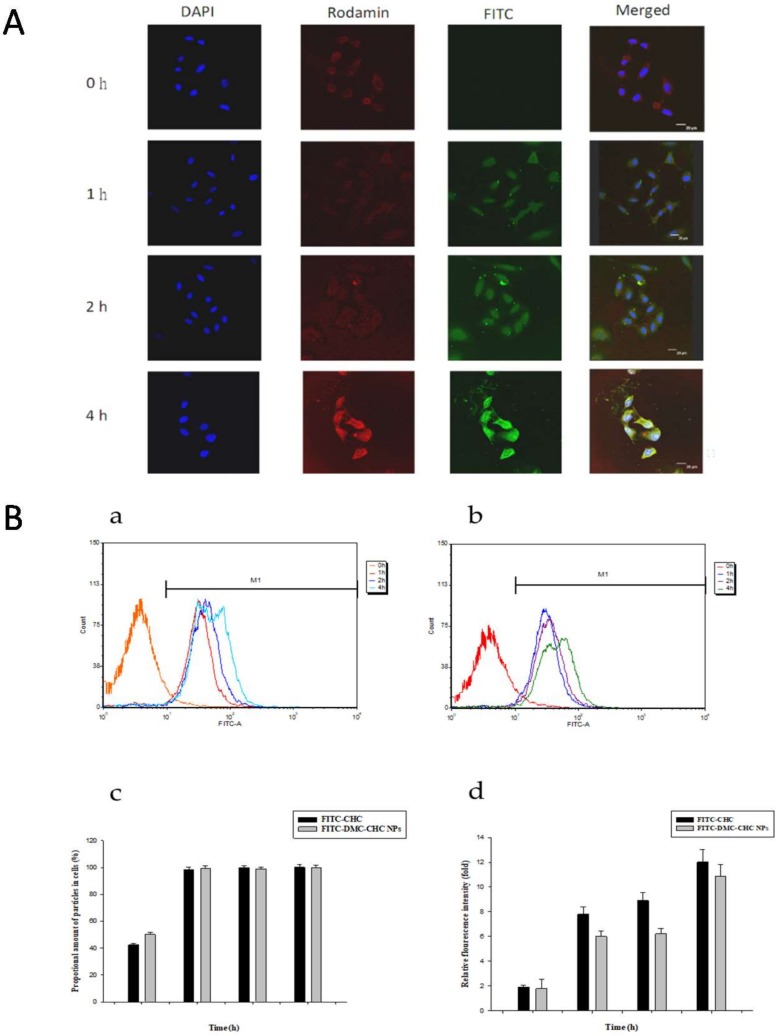
(**A**) Fluorescence images of A549 incubated with the DMC-CHC nanoparticles (5 μg/mL) for 0–4 h. The merged image was the overlapping image obtained using the DAPI channel (blue, nuclei), rhodamine channel (red, F-actin), and FITC channel (green, nanoparticles). (B) Nanoparticles uptake efficiency measured by flow cytometer (**a**) 20 μg/mL FITC-CHC and (**b**) 5 μg/mL FITC-DMC-CHC NPs in A549. A549 were administered with or without FITC-DMC-CHC NPs for various time intervals (0–4 h). (**c**) To compare the proportional amount of nanoparticles, and (**d**) relative fluorescence intensity between FITC-CHC and FITC-DMC-CHC NPs in A549 cancer cells (M1). 10,000 cells were randomly chosen for imaging analysis with flow cytometer. Each data point is represented as the mean ± SD (*n* = 3). Error bars represent the standard deviation.

**Figure 3 molecules-23-03217-f003:**
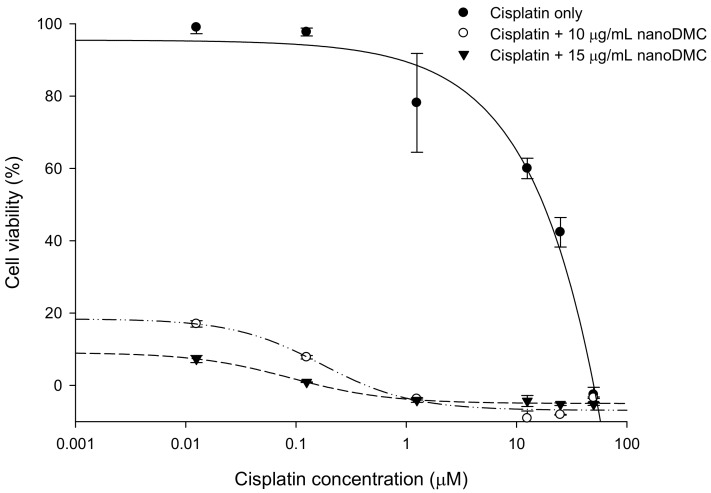
Effects of DMC-CHC NPs on cytotoxicity of cisplatin in A549 cells.

**Figure 4 molecules-23-03217-f004:**
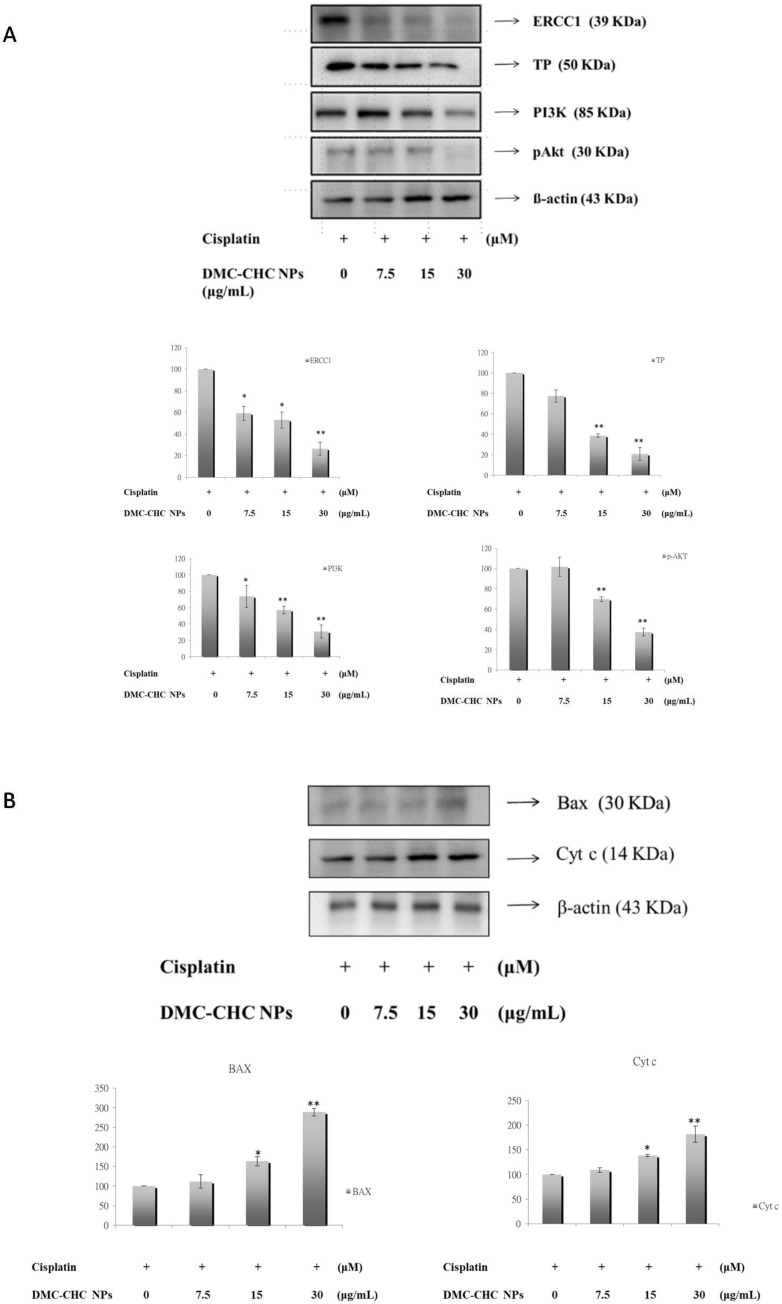
An impact from DMC-CHC NPs on on- and post-target resistance of CDDP was examined. (**A**) The effects of DMC-CHC NPs on on-target resistance of cisplatin was determined. DMC-CHC NPs significantly decreased cisplatin-induced TP and PI3k/Akt/ERCC1 signaling in A549 cell. A549 cells were exposed to various concentrations of DMC-CHC NPs (7.5 µg/mL, 15 µg/mL, and 30 µg/mL) and cisplatin for 48 h. (**B**) The effects of DMC-CHC NPs (7.5 µg/mL, 15 µg/mL, and 30 µg/mL) on post-target resistance of cisplatin was examined. The protein expression of bax, and cytochrome *c* in A549 cells were determined using Western blotting analysis. The results represent the mean ± SD of three independent experiments. * Indicates the values significantly different from the control. (* *p* < 0.05; ** *p* < 0.01).

**Table 1 molecules-23-03217-t001:** Characterizations of DMC-CHC NPs. One standard deviation is given within parentheses (*n* = 3).

Sample	D_h_ (nm)	Zeta Potential (mV)
CHC NPs	50 (10)	−22 (0.65)
DMC-CHC NPs	130 (21)	−9 (1.1)

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
