# Peer review of "Demethoxycurcumin-Loaded Chitosan Nanoparticle Downregulates DNA Repair Pathway to Improve Cisplatin-Induced Apoptosis in Non-Small Cell Lung Cancer"

_molecules, 2018, doi:10.3390/molecules23123217_

Reviewer 1 Report

The authors showed that carboxymethyl-hexanoyl-chitosan (CHC) nanoparticles are an effective drug delivery carrier of demethoxycurcumin (DMC) to lung cancer cells (A549); there is a relatively rapid penetration of nanoparticles into the cells and an effective overcoming of cisplatin resistance. Moreover, the authors experimentally showed that this effect is achieved by suppressing the expression of specific proteins responsible for cisplatin resistance. Generally, the paper is well written and seems scientifically competent within my areas of expertise. In my opinion, the manuscript can be considered for publication in Molecules after addressing the following comments.

1. Section 2.1: It would be feasible to add characteristics of unloaded CHC nanoparticles (such as hydrodynamic radius, zeta-potential, TEM/SEM morphology), and to compare with of DMC-CHC nanoparticles.

2. Section 2.1: It would also be helpful to provide the drug encapsulation efficiency: DEE (%) = [(mass of DMC in nanoparticles)/(mass of DMC-CHC nanoparticles)].

3. Lines 96-97: You may remove this sentence because it is duplicated in Materials and Methods (lines 289-290).

4. Figure 1B. Please, indicate what is marked by asterisks. You also mixed up left and right panels.

5. Section 2.4, Figure 2A: Why is the fluorescence intensity of the rhodamine channel significantly higher after 4 h incubation? Shouldn't it be the same?

6. Materials and Methods: All the polymers used in the study must be thoroughly characterized. Please provide the source, the molecular weight (MW) and the degree of deacetylation (DDA) of chitosan along with MW, DDA and the degrees of substitution (DS) for both functional groups (carbomethyl- and hexanoyl) of CHC. Provide also the methods (with appropriate references) used to determine these characteristics.

7. Materials and Methods: The description of TEM experiments is missed.

8. Conclusion: Remove Figure 5 because it is duplicated information from other Figures. You may consider using Figure 5 as a graphical abstract.

9. The quality of some figures (e.g., Figure 1B, 2B) is quite weak.

10. The previous paper (Huang, W.-T., Larsson, M., Wang, Y.-J., Chiou, S.-H., Lin, H.-Y., & Liu, D.-M. (2015). Demethoxycurcumin-Carrying Chitosan – Antibody Core-Shell Nanoparticles with Multitherapeutic Efficacy by Malignant A549: From in Vitro Characterization to Vivo Evaluation. Molecular Pharmaceutics, 12 (4), 1242–1249. doi: 10.1021 / mp500747w) the authors showed that the release of DMC from DMC-CHC nanoparticles is relatively slow. For 48 and 72 h incubation used in the current work, about 30% of DMC is released. I would suggest that the authors comment on how the observed effects would behave in time (in the release time scale shown in the previous work).

Author Response

Dear Professor/Reviewer:

I would like to take this chance to thank you for the valuable suggestions. The followings you will find the corresponding answers.

The authors showed that carboxymethyl-hexanoyl-chitosan (CHC) nanoparticles are an effective drug delivery carrier of demethoxycurcumin (DMC) to lung cancer cells (A549); there is a relatively rapid penetration of nanoparticles into the cells and an effective overcoming of cisplatin resistance. Moreover, the authors experimentally showed that this effect is achieved by suppressing the expression of specific proteins responsible for cisplatin resistance. Generally, the paper is well written and seems scientifically competent within my areas of expertise. In my opinion, the manuscript can be considered for publication in Molecules after addressing the following comments.

1. Section 2.1: It would be feasible to add characteristics of unloaded CHC nanoparticles (such as hydrodynamic radius, zeta-potential, TEM/SEM morphology), and to compare with of DMC-CHC nanoparticles.

[Answer]: please see the attached figure.

Figure. TEM morphology. Left one is unloaded CHC, and right one is DMC-CHC nanoparticles

Reference: Demethoxycurcumin-Carrying Chitosan–Antibody Core–Shell Nanoparticles with Multitherapeutic Efficacy toward Malignant A549 Lung Tumor: From in Vitro Characterization to in Vivo Evaluation, Wei-Ting Huang, Mikael Larsson, Yen-Jen Wang, Shih-Hwa Chiou, Hui-Yi Lin, and Dean-Mo Liu, Molecular Pharmaceutics 2015 12 (4), 1242-1249, DOI: 10.1021/mp500747w

Table. Hydrodynamic diameter from DLS and zeta potential of different formulations in pH 7.4 PBS. One standard deviation is given within parentheses (n = 3).

Sample

D (nm)

Zeta Potential (mV)

CHC

50 (10)

-22 (0.65)

CHC/DMC

130 (21)

-9 (1.1)

Reference: Wei-Ting Huang, Mikael Larsson, Yi-Chi Lee, Dean-Mo Liu, Guang-Yuh Chiou, Dual drug-loaded biofunctionalized amphiphilic chitosan nanoparticles: Enhanced synergy between cisplatin and demethoxycurcumin against multidrug-resistant stem-like lung cancer cells, European Journal of Pharmaceutics and Biopharmaceutics,2016, 165-173, https://doi.org/10.1016/j.ejpb.2016.10.014.

2. Section 2.1: It would also be helpful to provide the drug encapsulation efficiency: DEE (%) = [(mass of DMC in nanoparticles)/(mass of DMC-CHC nanoparticles)].

[Answer]: According to the above formula, the DEE (%) equals to 16.4%

3. Lines 96-97: You may remove this sentence because it is duplicated in Materials and Methods (lines 289-290).

[Answer]: Lines 96-97 has been deleted. Thanks for your suggestion.

4. Figure 1B. Please, indicate what is marked by asterisks. You also mixed up left and right panels.

[Answer]: This has been corrected as the following as shown in red pen: “Figure 1. (A) TEM images of DMC-CHC nanoparticles (scale bar: 200 nm). (B) Left panel: effect of DMC on the viability of A549 cells. Right panel: effect of free DMC-CHC NPs on the viability of A549 cells. (C) CHC nanomatrix (6.25, 12.5, 25, 50, and 100 μg/mL) were incubated with A549 for 24 h, 48 h. and 72 h. The results showed that the CHC nanomatrix demonstrated no cytotoxicity on A549. Each data point is represented as mean ± SD (n = 3). Error bars represent standard deviation. *Indicates the values significantly different from the control time points, 24 h, 48 h. and 72 h., respectively. *Indicates the values significantly different from the control. (*p < 0.05; ⁎⁎p < 0.01).”

5. Section 2.4, Figure 2A: Why is the fluorescence intensity of the rhodamine channel significantly higher after 4 h incubation? Shouldn't it be the same?

[Answer] thanks for your valuable suggestion. The fluorescence has been shown in a closer intensity.

6. Materials and Methods: All the polymers used in the study must be thoroughly characterized. Please provide the source, the molecular weight (MW) and the degree of deacetylation (DDA) of chitosan along with MW, DDA and the degrees of substitution (DS) for both functional groups (carbomethyl- and hexanoyl) of CHC. Provide also the methods (with appropriate references) used to determine these characteristics.

[Answer]: Thanks again. The characterizations of the ploymer used was added. An amphiphilic chemically-modified chitosan, i.e., carboxymethyl-hexanoyl chitosan (CHC), was successfully synthesized as previously described (Huang et al., 2015; Wang, Lin, Wu, & Liu, 2012) and was purchased from (Advanced Delivery Technologies, Inc. Taiwan) and is used without further purification. The CHC has following properties: deacylation of 91.7%, average molecular weight of 35,366 g/mole (with PDI 1.12), and degree of carboxymethylation of 55.3%. The average molecular weight was determined by GPC (TSKgel SuperAW4000 Column; PU-4180 RHPLC Pump, RI-4030 Refractive Index Detector, LC-NetII/ADC Interface Box and CO-4060 Column Oven) using polyethylene (PEO/PEG) as standards and the degree of carboxymethylation was determined by FT-IR.

Reference: Wei-Ting Huang, Yi-Ping Ko, Ting-Yu Kuo, Mikael Larsson, Min-Chih Chang, Ren-Der Jean, Dean-Mo Liu, A new type of gadodiamide-conjugated amphiphilic chitosan nanoparticle and its use for MR imaging with significantly enhanced contrastability, Carbohydrate Polymers, 2019, 256-264, 0144-8617, https://doi.org/10.1016/j.carbpol.2018.09.031.

7. Materials and Methods: The description of TEM experiments is missed.

[Answer]: The TEM has been added to the Materials and Methods section.

The structure of CHC and DMC-CHC nanoparticles were analyzed through transmission electron microscope (TEM, JEOL 2100, Japan). TEM samples were prepared immediately after CHC and CHC/DMC nanoparticles had been synthesized. Both samples (1 mg/mL) were put on 200-mesh copper grids coated with carbon. For the CHC/DMC nanoparticle sample, negative staining was used to observe DMC in CHC carrier.

Reference: Demethoxycurcumin-Carrying Chitosan–Antibody Core–Shell Nanoparticles with Multitherapeutic Efficacy toward Malignant A549 Lung Tumor: From in Vitro Characterization to in Vivo Evaluation, Wei-Ting Huang, Mikael Larsson, Yen-Jen Wang, Shih-Hwa Chiou, Hui-Yi Lin, and Dean-Mo Liu, Molecular Pharmaceutics 2015 12 (4), 1242-1249, DOI: 10.1021/mp500747w

8. Conclusion: Remove Figure 5 because it is duplicated information from other Figures. You may consider using Figure 5 as a graphical abstract.

9. The quality of some figures (e.g., Figure 1B, 2B) is quite weak.

[Answer]: The quality of Figures 1B and 2B have been intensified.

10. The previous paper (Huang, W.-T., Larsson, M., Wang, Y.-J., Chiou, S.-H., Lin, H.-Y., & Liu, D.-M. (2015). Demethoxycurcumin-Carrying Chitosan – Antibody Core-Shell Nanoparticles with Multitherapeutic Efficacy by Malignant A549: From in Vitro Characterization to Vivo Evaluation. Molecular Pharmaceutics, 12 (4), 1242–1249. doi: 10.1021 / mp500747w) the authors showed that the release of DMC from DMC-CHC nanoparticles is relatively slow. For 48 and 72 h incubation used in the current work, about 30% of DMC is released. I would suggest that the authors comment on how the observed effects would behave in time (in the release time scale shown in the previous work).

[Answer]: Thanks for your suggestion. The following description has been added to the second paragraph in the Discussion section as following: “In our previous study, the data showed that free DMC was completely eluted in 24 h, while DMC-CHC NPs showed a sustained, slow profile for a time period of 10 days (corresponding to an accumulative amount of about 65%). In the present study, for 48 h. and 72 h. incubation period, this evidence afford the availability that the DMC, gradually slow-released from the nanoparticle, could effectively work on the cisplatin-induced resistance”

Reviewer 2 Report

The summary shows the scope of the work developed by the group. It is important to include the novelty and contribution of this study presented in a clear and punctual form.
The study of the state of the art of the study area is shown. Adequate and updated literature is displayed. The study indicates that NER plays an important role and the role of different compounds with respect to cancer cells. The authors analyse the literature and identify the various contributions of the various research groups. And based on this review of the literature identify and describe the contribution to this proposal. In the same way, the Working Group has recent studies, which indicates the experience in the area of knowledge. Therefore, they propose the controlled uses of DMC, as a possibility for the DMC-CHC NPs to boost cisplatin chemotherapy, and its correlation with the signaling pathways TP and ERCC1. The literature has references in the subject of research, so it is recommended to strengthen the novelty and relevance of this study.

The design of the research and methodology are described in an appropriate way. The results require in some sections, a greater discussion and analysis supported with scientific literature. See comments in PDF.

The conclusion requires better. The description is general and does not seem a conclusion. This has to be defined because the materials studied worked for the application.

Author Response

Dear Professor/Reviewer:

Thank you so much for your valuable suggestions. All of the response has been shown on the attached file.

Round  2

Reviewer 1 Report

The authors have successfully addressed most of the reviewer’s concerns, improving the manuscript with their edits. However, there are still some minor points for authors to consider:

Line 18: Replace ‘hexanol’ to ‘hexanoyl’

Line 105: Add (Dh) after ‘radius’

Table 1: Replace D to Dh, CHC to CHC NPs, CHC/DMC to DMC-CHC NPs

Lines 130,132: Replace CHC/DMC to DMC-CHC

Line 142: Remove duplicated ‘*Indicates the values significantly different from the control.’

Line 321: Molecular weight is unitless, please remove g/mole’. Round off the molecular weight to two (max. three) significant figures as 3.5×104

Line 321: Replace ‘deacylation’ to ‘degree of deacetylation’

Line 322: Replace ‘PDI’ to ‘dispersity of’

Line 340: Add × 100 at the end